# No Sex-Specific Effects of Artificial Selection for Relative Telencephalon Size during Detour Learning and Spatial Discrimination in Guppies (*Poecilia reticulata*)

Annika Boussard *, Stephanie Edlund, Stephanie Fong, David Wheatcroft and Niclas Kolm

Department of Zoology/Ethology, Stockholm University, Svante Arrhenius Väg 18B,
SE-10691 Stockholm, Sweden; stephanie@djursmart.se (S.E.); stephanie.fong@zoologi.su.se (S.F.);
david.wheatcroft@zoologi.su.se (D.W.); niclas.kolm@zoologi.su.se (N.K.)
* Correspondence: annika.boussard@zoologi.su.se

**Abstract:** Over recent decades, substantial research has focused on fish cognitive evolution to increase our understanding of the evolution of the enormous diversity of cognitive abilities that exists in fishes. One important but understudied aspect of cognitive evolution is sexual dimorphism in cognitive abilities. Sex-specific variation in brain region morphology has been proposed to be an important mechanism in this context. However, it is also common to find sex-specific variation in behavior and cognition without associated differences in brain morphology among the sexes. The telencephalon is the major cognitive center in the vertebrate brain and variation in telencephalon size has been associated with variation in cognition. Here, we utilize recently developed guppy artificial selection lines with ca. 10% differences in relative telencephalon size to investigate whether similar responses to selection of the size of this region may affect cognitive abilities differently in males and females. To that end, we compared two ecologically relevant aspects of cognition, detour learning and binary spatial discrimination. We tested the significance of the interaction between telencephalon size and sex, and we found no sex-specific effects of evolutionary increases in telencephalon size in the cognitive abilities tested. This study indicates that no clear cognitive sex-specific effects occur in response to rapid selection of telencephalon size. We suggest that future research on sexual dimorphism in cognitive abilities in fish could use various cognitive tests and examine telencephalic sub-regions to gain a more comprehensive understanding of their evolution.

**Keywords:** detour learning; spatial discrimination; telencephalon; cognitive sexual dimorphism

**Key Contribution:** This is the first experimental test to assay both males and females to investigate the cognitive consequences of rapid evolutionary changes in artificial selection lines with differences in relative brain region size. We found no sex-specific effects of artificial selection of relative telencephalon size on detour learning ability or binary spatial discrimination.



## 1. Introduction

How fishes acquire, process, store and act on environmental cues has received an increasing amount of research during the recent decades [1–6]. As fishes represent the ancestral state of all vertebrates, investigating how and why cognitive abilities differ across fish species may offer insights into the evolutionary history of cognitive abilities in all vertebrates.

Various socio-ecological challenges can cause divergent selection pressures [7–15]. Such challenges are often sex-specific, due to reproductive systems and sex-specific roles. Cognitive sexual dimorphism is, therefore, important to consider as it may have implications for cognitive evolution, especially when cognitive selection pressures differ between the sexes. Cognitive sexual dimorphism can correspond to matching size differences in

brain structures [9,16–18]. For instance, male sticklebacks invest more into cognitively demanding reproductive behaviors, such as parental care and building nests, and have larger brains than their female conspecifics [18]. Furthermore, an artificial selection experiment on male genital length in eastern mosquitofish suggested a positive genetic correlation between male gonopodium length and female brain size [19]. The authors argued that this is likely caused by the increased female cognitive abilities required to avoid male coercion. Since the telencephalon is the cognitive center in fish [13,20–22], this is particularly interesting in the context of fish cognitive evolution. In fish, telencephalon size is positively correlated with habitat complexity [9,16,23], social complexity [14], and executive functions, such as object permanence and detour learning, in laboratory-bred guppy lines [24–26]. Furthermore, several comparative and experimental studies have linked cognitive sexual dimorphism to sexual dimorphism in telencephalon size and its neural structures [9,16,19,27].

Sex-specific differences in cognitive abilities cannot solely be attributed to macrostructural differences in brain regions. This is exemplified by studies on Eastern mosquitofish (*Gambusia holbrooki*) and Cocos frillgoby (*Bathygobius cocosensis*). It has been reported that cognitive abilities, processed in the telencephalon, differ between the sexes [7,8,13,19,23] while relative telencephalon sizes remain consistent [19,23]. Moreover, the well-documented sex-specific cognitive differences in laboratory-bred guppies (e.g., [28–31]), with the lack of disparity in relative telencephalon size between the sexes under controlled conditions [32], further corroborate this statement. The study of sexual dimorphism in cognition among fish with comparable telencephalon sizes in both males and females can thus advance our understanding of cognitive evolution.

In the guppy, predation pressure, mating strategies and social behaviors vary greatly between the sexes [8,33]. Standard cognitive tests have shown sexual dimorphism in several aspects of cognition. Female guppies outperform males in, e.g., serial reversal learning [28], problem-solving [30] and motor self-regulation during foraging [31], whereas males show increased spatial maze navigation abilities [29], and cylinder detour learning abilities [30]. The sex-specific socio-ecological challenges and the cognitive sexual dimorphism in the guppy makes it an ideal model to test a hypothesis on the concept that males and females differ in cognition and underlying brain structures.

In this study, we investigate how artificial selection of relative telencephalon size is related to cognitive ability in male and female guppies. To do so, we assay two aspects of cognitive ability, detour learning and binary spatial discrimination. We use guppies artificially selected for relative telencephalon size during four generations of selection with approximately 10% difference in both sexes in telencephalon volume between up- and down-selected lines (see details of the selection lines below). The selection for relative telencephalon size so far shows very similar patterns in the neural response to selection between the sexes in the guppy telencephalon lines [32]. However, guppies artificially selected for relative brain size have no sexual dimorphism in relative brain size [34]. Yet, in those selection lines, there are several sex-specific differences in behavior. For instance, these have been noted in standard cognitive tests [34] and predator avoidance experiments [10,35]. Therefore, examining potential sex-specific effects of artificial selection on relative telencephalon size in cognitive abilities could be relevant to understand how cognition evolves in these recently developed telencephalon size selection lines. Since cognitive sexual dimorphism in detour and spatial abilities is well-established in guppies (e.g., [8,11,27–30], only sex-specific effects of telencephalon size selection in detour and spatial learning will be tested and discussed.

## 2. Methods

### 2.1. The Guppy Telencephalon Size Selection Lines

We conducted this study between October and December 2020 at the Stockholm University Zoology Department fish laboratory facilities. Descendants of wild-caught guppies from high-predation areas in the Quare River in Trinidad were used as the starting population for the telencephalon artificial selection lines. Three independent breeding stocks

(hereafter replicates) with 75 breeding pairs each were set up as the $F_0$ generation. In these three replicates, offspring from breeding pairs with the 20% largest relative telencephalon size, the 20% smallest relative telencephalon size in relation to the rest of the brain and randomly chosen controls were used to create three up-selected, three down-selected and three control lines. Relative telencephalon size was established through quantification of the residuals of telencephalon volume on the volume of the rest of the brain. The whole brain was dissected from euthanized fish, and the length, width and height were measured. The volume was determined using the ellipsoid model described in ref [12,36]. Juveniles were separated from their parents at birth, and once sexually mature, fish were divided by sex and housed into groups of 8–9 females and 10–12 males in 7l holding tanks, with 2 cm gravel, three snails (*Planorbis* sp.), java moss (*Taxiphyllum* sp.), constant aeration and $25 \pm 1$ °C water temperature, on a 12:12 dark: light scheme. Fish were fed six days per week with flake food and *Artemia* hatchlings. In the present study, fish were collected from random holding tanks; only fish from the up- and down-selected lines were included. We used 60 individuals, equally distributed between sex, selection line and replicates, from the 4th generation, which had approximately 10% difference in relative telencephalon size between the up- and down-selected lines in both sexes [32]. This sample size was no smaller than sample sizes in previous studies using male and female guppies [28–31]. Our sample size should thus have had sufficient power to detect effects. For more details of the artificial selection experiment on relative telencephalon size, see ref [32].

### 2.2. Detour Learning Test

To test detour learning ability, we used a cylinder task [37]. All fish were kept in individual 7l experimental tanks, enriched with 2 cm white gravel and plastic plants, throughout the experiment. The experimental room was maintained under equal light, aeration and temperature conditions as the main laboratory facilities. The experimental tanks consisted of a home compartment separated from an experimental chamber by an opaque sliding door (Figure 1a). Visual contact was maintained between the different experimental tanks' home compartments in order to avoid any potential negative effects of social isolation, but visual contact was prevented between the experimental chambers to avoid social learning [38]. In the experimental chamber, a white plate (10 × 15 cm) with a green mark (2 mm in diameter) in the center was placed on the bottom. Prior to the detour learning task, the guppies were fed with adult thawed *Artemia* on the green mark twice per day for five consecutive days. All subjects learnt to find the *Artemia* placed on this green mark within this period. On the sixth day, a transparent cylinder (4 cm in diameter × 5 cm long) was positioned horizontally on top of the green mark, with the opening ends not directly in view from the sliding door. The green mark and the *Artemia* reward were clearly visible through the cylinder. In order to obtain the reward, the guppies now had to detour around to one of the open ends of the cylinder. The guppies had no prior experience with an opaque cylinder, to avoid measuring cognitive processes unrelated to the motoric self-regulation required to learn to detour around an obstacle [39,40]. The guppies were trained over 12 consecutive days for a total of 40 trials. The number of trials per day increased from two trials on the first day to five trials on the last day, as the subjects solved the task at a faster rate towards the last days of training. Detour learning was quantified as the latency (s) from entering the experimental chamber to when they obtained the *Artemia* as well as the number of attacks on the cylinder (i.e., touching or bouncing into the cylinder with their snout, following ref. [30]. A trial ended when the *Artemia* reward was obtained. The guppies had access to the home compartment at all times, but they only had access to the experimental chamber during pre-training to feed on the green mark and during trials. Two observers (AB and SE) scored the behavior of each fish. To minimize observer bias, a person unrelated to the experiment transferred the fish from holding tanks to the experimental tanks that were only identifiable by running numbers.

**(a)**

**(b)**

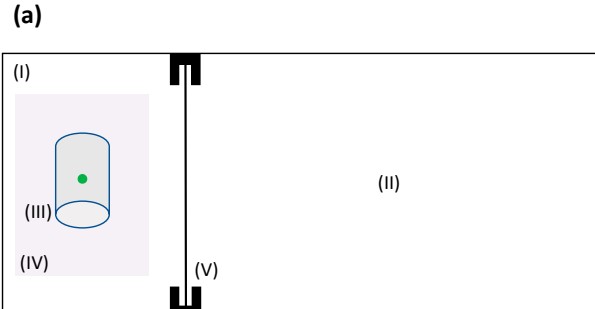
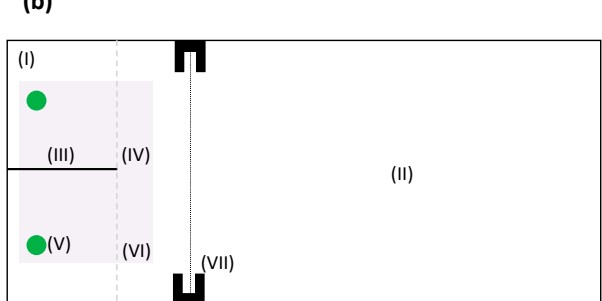

**Figure 1.** Top views of the experimental set-ups used in the two cognitive assays. Both set-ups consisted of an experimental chamber (I) and a home compartment (II). (**a**) In the detour learning set-up, a transparent cylinder (III) was positioned horizontally on top of a green mark painted on a white plate (IV) placed on the bottom. The experimental compartment and home compartment were separated by an opaque sliding door (V). (**b**) In the spatial discrimination set-up, an opaque plastic divider (III) was placed on a white plate (IV) on the bottom in the experimental chamber. On each side of the divider, a green plastic disc (V) was placed. When the subject crossed a line (VI), this behavior was scored as correct, and the subject was then rewarded on the green disc, or it was scored as an error depending on which side of the divider the subject was trained to associate with food. The experimental compartment and home compartment were separated by a transparent sliding door (VII).

## 2.3. Binary Spatial Discrimination Test

The binary spatial discrimination test followed after the detour task for all individual subjects (one up-selected male died prior to this test, i.e., n = 59). This time, the guppies were initially fed an adult thawed *Artemia* four times in a row on a green plastic disc placed in the middle on a white plate. This procedure enabled the guppies to associate a green disc with a food reward, which would facilitate the foraging behavior required during the binary spatial discrimination training. Following the initial training to associate a green disc with a food reward, a green disc was placed in the right-hand and left-hand sides of the experimental chamber (Figure 1b). These were separated by an opaque plastic divider. During a pre-trial, the first side the guppies swam to was noted and each individual guppy was then rewarded and trained against their initially preferred side, i.e., an individual that chose the left-hand side during the pre-trial was rewarded when changing to the right-hand side, and thereafter trained on the right-hand side. This was performed to minimize the chances that the learning task would follow pre-existing individual side-biases [41]. An *Artemia* reward was delivered with a pipette on the green disc when the guppy crossed a line between the divider and the tank wall (see Figure 1b). The guppies were given a session consisting of four trials per day during the experiment. For each trial, we noted down if the guppy made a correct or incorrect choice, i.e., crossing the line. Training continued until an individual learning criterion of seven out of eight correct responses was reached, for a maximum of 100 trials (25 days). This learning criterion was chosen since it differs significantly from chance (binomial test; $p = 0.03$). The guppies had access to the home compartment at all times, but only access to the experimental chamber during trials. A single observer (AB) scored the behavior of each fish. Again, the experimental tanks were labeled with running numbers only in order to minimize observer bias. The true identities of the fish were accessible after the experiments.

## 2.4. Statistical Analyses

All analyses were performed and all figures were generated in R statistical software (v 4.3.1, http://R-project.org/, accessed on 16 June 2023).

### 2.4.1. Detour Learning

To examine sex-specific effects of relative telencephalon size on detour learning ability, we assessed latency and number of attacks into the cylinder (i.e., the number of times they touched the cylinder with their snout) until the task was completed (i.e., until the *Artemia* reward was consumed). For the model with latency as the dependent variable, variance differed significantly between the selected lines (Fligner–Killeen; $\chi^2_1 = 5.93$, $p = 0.01$). Therefore, we fitted a generalized least square model (GLS) as implemented in the *gls* function in the *nlme* package [42]. Latency was log-transformed to better meet the assumptions of the model. For the model with number of attacks as the dependent variable, we ran a generalized linear mixed-effects model (GLMM) with logit link functions as implemented in the *glmer* functions in the *lme4* package [43]. Since the variance increased quadratically with the mean, the number of attacks was modeled under a type II negative binomial distribution. We modeled both dependent variables as a function of the explanatory variables telencephalon size, sex, trial number and their two-way interactions. We also included a three-way interaction between sex, telencephalon size and trial number. Fish ID was included as a random intercept and slope to account for repeated measurements and individual variation in learning slopes [44]. Prior to obtaining model fit, the explanatory variable 'trial' was standardized by mean centering and divided by 1 standard deviation as this improves the numerical optimization process of linear models. The observer was not accounted for in the models, as a single observer scored all behaviors in the first half of the experiment and the other observer in the second part of the experiment.

### 2.4.2. Spatial Discrimination

To examine the sex-specific effect of telencephalon size in binary spatial discrimination ability, we modeled this relationship as proportion count data (i.e., R glmer syntax cbind (correct, errors)) under a binomial assumption. We ran a GLMM with logit link functions. We modeled the dependent variable as a function of the explanatory variables telencephalon size and sex, and an interaction between these terms, to investigate if potential differences in slopes between the selected lines changed with sex. Since we had multiple observations per subject, fish ID was included as a random intercept to account for repeated measurements.

Since the number of trials was not independent with the same probability within each subject, we computed an approximate estimate of an overdispersion factor. Briefly, we did this by dividing the sum of squared Pearson residuals with the residual degrees of freedom. There was no evidence for overdispersion of the model ($\chi^2 = 16.4$, ratio = 0.32, rdf = 52, $p = 1$).

Telencephalon size was treated as a two-level categorial variable (small, large) in the models described above. Initially, a random intercept was included for telencephalon size nested in replicates to account for potential variation between the replicated lines and selection regime. In cases where replicates returned a zero-variance estimate (i.e., caused singular fit to the model), we used a two-step approach following ref. [45]. First, we simplified the random structure to include only a random intercept for replicates. If replicates still returned a zero-variance estimate, we fitted replicates as a fixed factor, but excluded it from further analyses if the fit of the models did not improve significantly ($\Delta$AIC > 2). Once the model was fitted, model validation was applied in which Pearson residuals were plotted versus fitted values and visually inspected for heteroscedasticity. Normality of residuals was confirmed via visual inspection. Since the aim of the study was to examine sex-specific effects of telencephalon size selection, only the interaction terms of interest were interpreted. Therefore, no model was simplified. Test statistics and *p*-values were obtained by using the ANOVA function, specifying type III Wald chi-square tests, in the *car* package [46].

## 3. Results

### 3.1. Detour Learning Assay

We found no sex-specific effect of artificial selection on relative telencephalon size in latency to detour, as the three-way interaction between telencephalon size, trial and sex was not significant (GLS; $\chi^2_1 = 1.66$, $p = 0.20$; Figure 2a,b). No individual learnt to detour correctly in the strictest sense of the term (no touching of the cylinder after learning to detour correctly). As for latency, we found no interaction effect between telencephalon size, trial and sex in the number of attacks (GLMM; telencephalon size × trial × sex: $\chi^2_1 = 1.96$, $p = 0.16$; Figure 2c,d). This means that we found no sex-specific effects of telencephalon size selection in either latency to detour or number of attacks into the cylinder with an increasing number of trials.

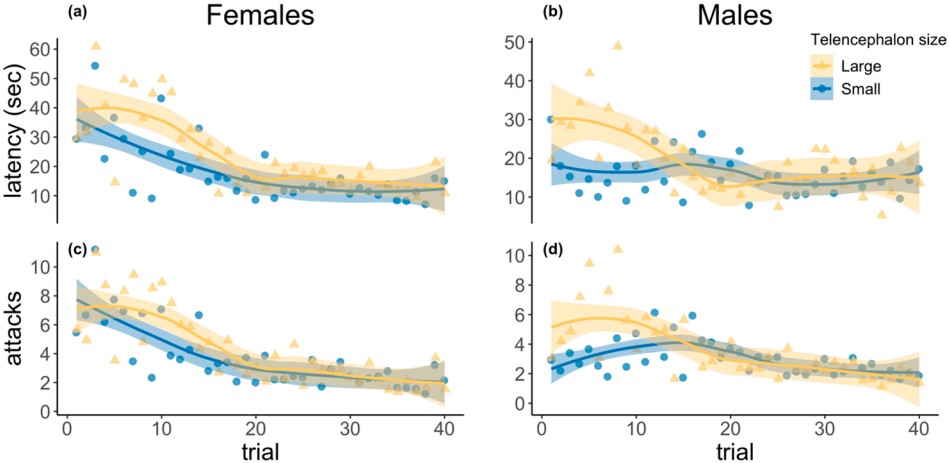

**Figure 2.** Detour learning assay. Learning curves over 40 trials for guppies artificially selected for large (yellow) and small (blue) relative telencephalon size. Top panels show latency in seconds until detour completed, in (**a**) females and (**b**) males, and lower panels show number of attacks (i.e., touched a cylinder with the snout), in (**c**) females and (**d**) males. Data points show raw data mean per trial. Solid lines show the smoothed conditional means with 95% confidence intervals (shading) obtained using a *gam* model. Data from 60 individuals.

### 3.2. Binary Spatial Discrimination Test

In the spatial binary discrimination test, the individual learning criterion (seven out of eight correct responses) was reached by all but sixteen individuals (five up- and eleven down-selected individuals). Males with a larger telencephalon made fewer errors (nearly half as many) before reaching the learning criterion compared to males with a smaller telencephalon (mean ± s.e.: $25.7 ± 7.4_{\text{up-selected lines}}$ vs. $48.4 ± 7.5_{\text{down-selected lines}}$; Figure 3). This pattern was not observable in females (mean ± s.e.: $27.9 ± 6.5_{\text{up-selected lines}}$ vs. $31.5 ± 7.5_{\text{down-selected lines}}$; Figure 3). However, we found no interaction effect between telencephalon size and sex on the proportion of correct choices until they reached a learning criterion (GLMM; telencephalon size × sex: $\chi^2_1 = 2.03$, $p = 0.15$, Table 1). Hence, overall, we found no sex-specific effect of artificial selection on telencephalon size in the binary spatial discrimination test.

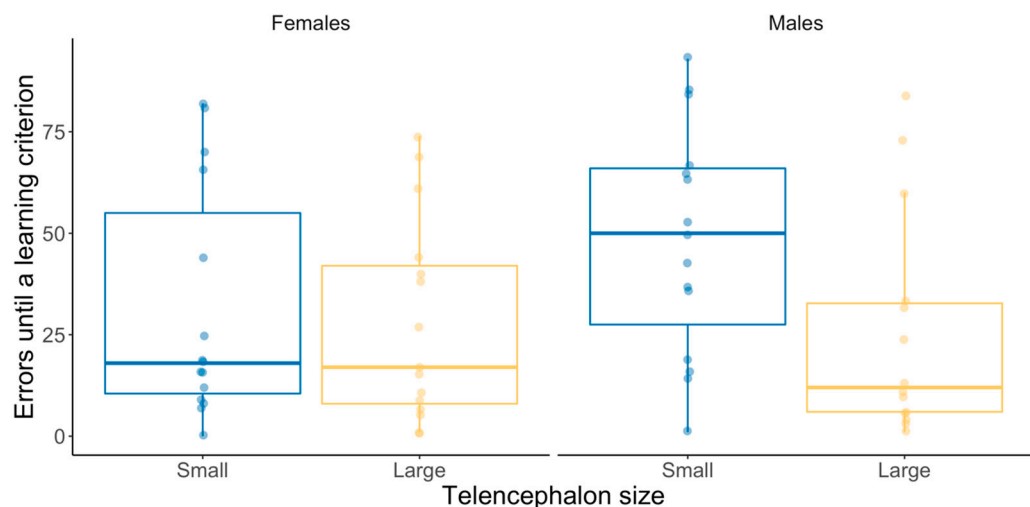

**Figure 3.** Binary spatial discrimination test. Boxplots of number of errors until reaching an individual learning criterion in male and female guppies from telencephalon size artificial selection lines. Boxplots indicate the median value and interquartile range. Data from 59 individuals.

**Table 1.** Results from a GLS and two GLMMs testing sex-specific effects of artificial selection on relative telencephalon size in two cognitive assays. The columns provide chi-squared values ($\chi^2$), degrees of freedom (d.f.) and associated significance values (*p*) for the fixed explanatory variables included in each model. Significant ($p < 0.05$) effects are highlighted in bold. Reported also are the regression slope estimates and their standard errors (SEs). Telencephalon size, small; sex, female; and mid trial (see Section 2) are set as baseline.

| **Detour Learning Test** | | | | |
| --- | --- | --- | --- | --- |
| **Latency (GLS)** | $\chi^2$ | **d.f.** | **$p$-Value** | **Estimate $\pm$ SE** |
| (Intercept) | 410.40 | 1 | <0.001 | 2.34 (0.12) |
| Telencephalon size | 2.53 | 1 | 0.11 | 0.26 (0.16) |
| Trial | 40.44 | 1 | **<0.001** | −2.26 (0.04) |
| Sex | 0.55 | 1 | 0.46 | −0.12 (0.16) |
| Telencephalon size × Trial | 0.16 | 1 | 0.69 | 0.02 (0.06) |
| Telencephalon size × Sex | 0.20 | 1 | 0.65 | −0.10 (0.23) |
| Trial × Sex | 27.32 | 1 | **<0.001** | 0.30 (0.06) |
| Telencephalon size × Trial × Sex | 1.66 | 1 | 0.20 | −0.71 (0.08) |
| **Number of attacks (GLMM)** | | | | |
| (Intercept) | 179.08 | 1 | <0.001 | 1.15 (0.09) |
| Telencephalon size | 1.15 | 1 | 0.28 | 0.13 (0.12) |
| Trial | 55.23 | 1 | **<0.001** | −0.42 (0.06) |
| Sex | 0.82 | 1 | 0.36 | −0.11 (0.12) |
| Telencephalon size × Trial | 0.38 | 1 | 0.54 | −0.05 (0.08) |
| Telencephalon size × Sex | 0.11 | 1 | 0.74 | −0.06 (0.17) |
| Trial × Sex | 11.95 | 1 | **<0.001** | 0.28 (0.08) |
| Telencephalon size × Trial × Sex | 1.96 | 1 | 0.16 | −0.16 (0.11) |
| **Binary spatial discrimination test** | | | | |
| Proportion correct vs. error (GLMM) | | | | |
| (Intercept) | 0.21 | 1 | 0.65 | −0.09 (0.20) |
| Telencephalon size | 0.13 | 1 | 0.72 | 0.10 (0.29) |
| Sex | 3.17 | 1 | 0.08 | −0.51 (0.28) |
| Telencephalon size × Sex | 2.03 | 1 | 0.15 | 0.59 (0.4) |

## 4. Discussion

This is the first experimental test to investigate the cognitive consequences of artificial selection in relative brain region size in both sexes. We found no sex-specific effects

of artificial selection of relative telencephalon size on detour learning ability or binary spatial discrimination.

In previous work, the difference in relative telencephalon size between up- and down-selected lines was close to 10% in both sexes after four generations of selection, and no differences in the size of any other brain regions were detected between the sexes [32]. The lack of sex-specific effects of relative telencephalon size in the cognitive aspects tested here supports this. Interestingly, sex-specific cognitive differences have been found in guppies artificially selected for relative brain size, with no difference in relative brain size between the sexes [10,34,35]. Another possibility is that the cognitive challenge in these assays was not sufficiently challenging to detect sex-specific differences between the telencephalon size selection lines. For instance, associative learning rate does not differ in guppies with variation in relative brain size, but performance in reversal learning, a more challenging task, does [47]. One potential confounding effect that could have played a role in our assay of binary spatial discrimination is differences between the sexes and/or the telencephalon size selection lines in lateralization. Although we controlled for this by training the fish against their initial side preference, it is possible that individuals differ in their degree of lateralization, which could affect the difficulty in the task. However, we think this is unlikely since any side biases have not been detected in previous studies using these telencephalon size selection lines [24–26]. In light of this, we cannot completely rule out that there may still be sex-specific effects on behavior in the telencephalon size selection lines, but the effects are most likely small in that case and additional behavioral assays are required to investigate this further. We suggest that such assays should be more cognitively challenging to be able to more efficiently discover small effects.

A sex-specific effect of artificial selection on cognitive abilities would have supported the possibility that sub-regional neural tissues within the telencephalon had changed between the sexes with artificial selection. Since we found no such effect, this suggests that the artificial selection of telencephalon size increased or decreased all sub-regional brain tissues within the telencephalon in our guppy lines. The nonsignificant three- and two-way interactions of interest here corroborate this statement. This suggests that artificial selection of whole brain region size did not operate in a different manner between the sexes within our guppy population, at least not for areas processing detour learning and binary spatial discrimination. It is possible that in species with sex-specific differences in cognitive abilities, but similar relative telencephalon size, sub-regions within the telencephalon (or within other regions that do not display sex-specific size differences) may differ between the sexes. Such sex-specific differences in brain sub-regions have previously been demonstrated in a wild fish population. For instance, female blennies have a larger ventral subdivision of the area dorsalis telencephali lateralis than males, a pattern suggested to be driven by larger home ranges in females [16].

Sexual dimorphism in brain connectivity and neural density may also contribute to sex-specific differences in cognitive abilities [48–50]. While brain region size differences have historically been linked to cognitive differences between sexes [9,16–18], recent research highlights the critical roles of how neurons are interconnected and how densely they are packed [51–53]. Currently, we do not know if connectivity or neural density differ, in general, or in a sex-specific way, in the telencephalon size selection lines. But the present study suggests that if such effects exist, they did not affect the outcome in the cognitive assays. Regardless, future studies on sexual dimorphism in brain connectivity and neuronal density can offer a deeper understanding of the neural underpinnings of cognitive evolution in fish.

## 5. Conclusions

To conclude, this study suggests that rapid selection of relative telencephalon size does not yield distinct cognitive sex-specific responses. Future studies exploring potential sexual dimorphism in additional assays that test different cognitive abilities, and in telencephalic

sub-region sizes, could provide further insights into how and why sex-specific cognitive abilities evolve in fish.

**Author Contributions:** A.B. designed the assays, collected the data, carried out the statistical analyses and drafted the manuscript. S.F. created the telencephalon selection lines. S.E. collected the data. D.W. helped with the statistical analyses and writing the manuscript. N.K. conceived the idea for the telencephalon selection lines, created the telencephalon selection lines, designed the study and helped write the manuscript. All authors have read and agreed to the published version of the manuscript.

**Funding:** This work was funded by the Knut and Alice Wallenberg Foundation, Grant/Award Number: 102 2013.0072 (to N.K.), and the Swedish Research Council, Grant/Award Number: 2016-03435 (to N.K.).

**Institutional Review Board Statement:** The experiments were performed in accordance with the ethical conditions approved by the Stockholm Animal Research Ethical Permit Board (Dnr: N173/13, 223/15, N8/17 and 17362-2019).

**Informed Consent Statement:** Not applicable.

**Data Availability Statement:** The data and the R code to reproduce the statistical outcomes and generate the figures are available in the Figshare data repository: (https://figshare.com/articles/dataset/No_sex-specific_effects_of_artificial_selection_for_relative_telencephalon_size_in_detour_learning_and_spatial_discrimination_in_guppies_i_Poecilia_reticulata_i_/20452197, accessed on 23 October 2023).

**Acknowledgments:** We thank Vivien Holub and Eduardo Nila for fish housekeeping, Zegni Triki for discussions and Mirjam Amcoff who always lent a helping hand, along with the anonymous reviewers for their helpful comments on a previous version of the manuscript.

**Conflicts of Interest:** The authors declare no conflict of interest.

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
