# Peer review of "No Sex-Specific Effects of Artificial Selection for Relative Telencephalon Size during Detour Learning and Spatial Discrimination in Guppies (Poecilia reticulata)"

_fishes, doi:10.3390/fishes8110536_

Round 1
Reviewer 1 Report
Comments and Suggestions for Authors
This is an interesting study on intraspecific variation in cognition in a fish. I have few minor suggestions and a couple of general ideas that will hopefully help the authors to better present their study.
L52 Are those laboratory breed line artificially selected? If yes, I suggest to mention it explicitly.
L90 It is not clear whether there was a cognitive trait involved in the predator avoidance experiment. As it is reported now, it seems more a matter of shyness. Can you fix the wording or remove this citation if the effect is not attributable to cognition?
L108 three control lines?
L108-109 I think I know the procedure but other readers may not. Can you add a few lines to describe how the telencephalon size was estimated?
L145 in some cases, is it possible that the fish was simply contacting the cylinder while swimming towards the prey? As it is written it seems more of a deliberate attack towards the cylinder, which I don’t think was the behaviour observed. Please consider rewording or clarifications.
L162-167 I see the point of this procedure and it is notable that the authors tried to avoid a problem that is difficult to handle. However, this procedure has two drawbacks: 1) if individuals (or the two sexes) differ in the degree of lateralisation, than the task has different difficulty for them. 2) Additionally, the task became partially learning and partially reversal learning. I believe that there is no easy way out with all these issues of spatial tasks. My suggestion is only to acknowledge these potential confounding effects in discussion, just in case they played a role in the results.
L193 Usually the main variable for this task is the number (or proportion) of correct responses in which the fish does not touch the cylinder and directly reaches the food performing a ‘clean’ detour. See for instance the first paper in guppies on this task. This variable has a number of advantages that are certainly clear to the authors and it should be added to the analysis. It would also match the variables used in the following experiment (prop correct choices).
L248-250 I don’t grasp the meaning of this statement. Can it be better explained?
L285 The first sentence of discussion is a bit long, can it be shortned or simplified?
L290 ‘As mentioned’ can be removed. Also in similar statements below.
General ideas:
1) Is there a potential issue with power to be included in discussion? I see that there is a ‘marginal’ effect of sex (p = 0.08) in the spatial test, and indeed in guppies one of the study reported in introduction found a sex difference in a spatial task. Maybe, if the power is not sufficient to find a sex difference that is visible in the figure 3, it is quite unlikely sufficient to find a significant interaction between sex and selection line. To be on safe side this can be briefly mentioned, maybe arguing that the sample size is similar/smaller/larger to that of previous study.
2) Is it possible that the tasks are too easy to find this type of intraspecific differences? I mean differences between the sexes are typically small, here we are looking to a variation in a sex difference between selection lines. I am expecting it to be very small. Maybe it does not emerge if the test is not sufficiently challenging for the fish. I think is a cautionary interpretation to be mentioned in the discussion.
Author Response
Response to reviewer 1,
Thank you very much for taking the time to review this manuscript. Please find the detailed responses below and the corresponding revisions/corrections highlighted in the re-submitted file. We think your comments and ideas have improved the manuscript.
Point-by-point response:
Comment 1: L52 Are those laboratory breed line artificially selected? If yes, I suggest to mention it explicitly.
Response 1: Thanks for pointing this out. I have now changed the sentence to be more clear on that this is artificially selected lines.
Comment 2: L90 It is not clear whether there was a cognitive trait involved in the predator avoidance experiment. As it is reported now, it seems more a matter of shyness. Can you fix the wording or remove this citation if the effect is not attributable to cognition?
Response 2: I think you mean L96. Thanks for pointing this out. we agree, and have therefore fixed the wording to be clearer, with the following: “Yet, in those selection lines there are several sex-specific differences in behaviour. For instance, in standard cognitive tests (Kotrschal et al., 2013) and predator avoidance experiments (Kotrschal et al., 2015; van der Bijl et al., 2015).”
Comment 3: L108 three control lines?
Response 3: Yes, three control lines. It is added now.
Comment 4: L108-109 I think I know the procedure but other readers may not. Can you add a few lines to describe how the telencephalon size was estimated?
Response 4: Thanks for pointing this out. We have added a few sentences on line 118-120 to be clearer on how this was done, with the following: “The whole brain was dissected out from euthanized fish, and the length, width and height were measured. The volume was determined using the ellipsoid model described in Pollen et al., 2007, and White and Brown, 2015b.”
Comment 5: L145 in some cases, is it possible that the fish was simply contacting the cylinder while swimming towards the prey? As it is written it seems more of a deliberate attack towards the cylinder, which I don’t think was the behaviour observed. Please consider rewording or clarifications.
Response 5: We used the word "attacks" since it has been used previously by Lucon-Xiccato and colleagues. We have tried to clarify this further with the following: line 157-160: “ Detour learning was quantified as the latency (sec.) from entering the experimental chamber to when they obtained the Artemia as well as the number of attacks on the cylinder (i.e. touching or bouncing into the cylinder with their snout, sensu Lucon-Xiccato et al., 2020a)."
Comment 6: L162-167 I see the point of this procedure and it is notable that the authors tried to avoid a problem that is difficult to handle. However, this procedure has two drawbacks: 1) if individuals (or the two sexes) differ in the degree of lateralisation, than the task has different difficulty for them. 2) Additionally, the task became partially learning and partially reversal learning. I believe that there is no easy way out with all these issues of spatial tasks. My suggestion is only to acknowledge these potential confounding effects in discussion, just in case they played a role in the results.
Response 6: We agree. This is now mentioned in the second paragraph in the discussion. “One potential confounding effect that could have played a role in our assay of binary spatial discrimination, is differences between the sexes and/or the telencephalon size selection lines in lateralization. Although we controlled for this by training the fish against their initial side preference, it is possible that individuals differ in their degree of lateralization, which could affect the difficulty in the task. However, we think this is unlikely since any side biases have not been detected in previous studies using these telencephalon size selection lines (Triki et al., 2021; 2022; 2023).”
Comment 7: L193 Usually the main variable for this task is the number (or proportion) of correct responses in which the fish does not touch the cylinder and directly reaches the food performing a ‘clean’ detour. See for instance the first paper in guppies on this task. This variable has a number of advantages that are certainly clear to the authors and it should be added to the analysis. It would also match the variables used in the following experiment (prop correct choices).
Response 7: We totally agree, and we did indeed collect data on correct/successful detours (i.e., detouring without touching the cylinder). The problem here was that no guppy in this experiment learnt to detour correctly several times in a row. It only happened sporadically. Unfortunately, we do not have an sufficient amount of data to analyze it properly. We also clarified this in the results on line 253.
Comment 8: L248-250 I don’t grasp the meaning of this statement. Can it be better explained?
Response 8: Thanks for pointing this out. We hope it is clearer in the text now that we mean that there is no sex-specific effect.
Comment 9: L285 The first sentence of discussion is a bit long, can it be shortned or simplified?
Response 9: We agree, thanks for pointing this out. This sentence is shortened now with the following: “This is the first experimental test to investigate the cognitive consequences of artificial selection in relative brain region size in both sexes.”
Comment 10: L290 ‘As mentioned’ can be removed. Also in similar statements below. Response 10: This is changed now.
Comment 11:
1) Is there a potential issue with power to be included in discussion? I see that there is a ‘marginal’ effect of sex (p = 0.08) in the spatial test, and indeed in guppies one of the study reported in introduction found a sex difference in a spatial task. Maybe, if the power is not sufficient to find a sex difference that is visible in the figure 3, it is quite unlikely sufficient to find a significant interaction between sex and selection line. To be on safe side this can be briefly mentioned, maybe arguing that the sample size is similar/smaller/larger to that of previous study.
Response 11: Thanks for pointing this out. This is now mentioned in the methods, with the following: L129, “This sample size is not smaller than sample sizes in previous studies using male and female guppies (Lucon-Xiccato and Bisazza, 2014; Lucon-Xiccato et al., 2017; 2020a, 2020b). Our sample size should thus have sufficient power to detect effects.”
2) Is it possible that the tasks are too easy to find this type of intraspecific differences? I mean differences between the sexes are typically small, here we are looking to a variation in a sex difference between selection lines. I am expecting it to be very small. Maybe it does not emerge if the test is not sufficiently challenging for the fish. I think is a cautionary interpretation to be mentioned in the discussion.
Response: Good idea! We have added this to the second paragraph in the discussion. L310, “Another possibility is that the cognitive challenge in these assays was not sufficiently challenging to detect sex-specific differences between the telencephalon size selection lines. For instance, associative learning rate does not differ in guppies with variation in relative brain size, but performance in reversal learning, a more challenging task, does (Buechel et al., 2018).” And L324, “We suggest that such assays should be more cognitively challenging to be able to more efficiently discover small effects.”
Reviewer 2 Report
Comments and Suggestions for Authors
I have read through the manuscript entitled as ‘No sex-specific effects of artificial selection for relative telencephalon size in detour learning and spatial discrimination in guppies (Poecilia reticulata)’. I found the manuscript is interesting and well written. I only have some minor comments.
Abstract
Line 11 ’little’, line 13 ‘but’ and line 18 ‘sill’ consider revision.
Introduction
Line 39-40, the references should be listed either chronologically or alphabetically.
Line 45 Gonzalez-Voyer and Kolm, 2010;
Line 46 ‘in into’?
Line 52-53 consider revision
Line 85 we investigated
Methods
Line 116-116, why use 7l but two cm? please use consistent format
Line 190 ‘was placed a white plate’ consider revision
Results
Seems ok for me
Discussion
Line 296-298, delete the first Fong et al., 2021
Line 302-304, line 336-339 If possible, I would like to see the authors add something like these sentences in the abstract section.
Comments on the Quality of English LanguagePlease see my specific comments for authors listed aboved.
Author Response
Response to reviewer 2,
Thank you very much for taking the time to review this manuscript. Please find the detailed responses below and the corresponding revisions/corrections highlighted in the re-submitted file.
Point-by-point response:
Comment 1: Line 11 ’little’, line 13 ‘but’ and line 18 ‘sill’ consider revision.
Response 1: Thanks for pointing this out. We have now changed these words and deleted "still" in L18.
Comment 2: L39-40, the references should be listed either chronologically or alphabetically. Response 2: Pollen et al. have now changed place to fit in an alphabetically listed order.
Comment 3: L45 Gonzalez-Voyer and Kolm, 2010.
Response 3: Thanks for pointing these mistakes out. It should be corrected now.
Comment 4: L46 ‘in into’?
Response 4: Thanks for pointing these mistakes out. It should be corrected now.
Comment 5: L52-53 consider revision
Response 5: We agree, and have now rephrased this sentence, with the following: L54, “Since the telencephalon is the cognitive center in fish (Broglio et al., 2003, 2005; Portavella et al., 2002; Salas et al., 2003), it is particularly interesting in the context of fish cognitive evolution.”
Comment 6: L85 we investigated
Response 6: Respectfully, for tempus consistency throughout the paragraph we would like to keep “we investigate”.
Comment 7: L116-116, why use 7l but two cm? please use consistent format.
Response 7: We agree and have changed to 2cm to be consistent.
Comment 8: L190 ‘was placed a white plate’ consider revision.
Response 8: Corrected to "on a white plate".
Comment 9: L296-298, delete the first Fong et al., 2021.
Response 9: We agree, thanks for pointing this out.
Comment 10: L302-304, line 336-339 If possible, I would like to see the authors add something like these sentences in the abstract section.
Response 10: Thanks for pointing this out, but we think that it is already written in line 13-14, ”it is also common to find sex-specific variation in behaviour and cognition without associated differences in brain morphology among the sexes.”, but please let us know if we could be even clearer.
Line 336. Good point, I agree, and have now added this to the abstract on line 23-25, with the following “We suggest that future research on sexual dimorphism in cognitive abilities in fish could examine various cognitive tests and telencephalic subregions to gain a more comprehensive understanding of their evolution.”